# Security Aspects of Social Robots in Public Spaces: A Systematic Mapping Study

**DOI:** 10.3390/s23198056

**Published:** 2023-09-24

**Authors:** Samson Ogheneovo Oruma, Yonas Zewdu Ayele, Fabien Sechi, Hanne Rødsethol

**Affiliations:** 1Department of Computer Science and Communication, Østfold University College, 1757 Halden, Norway; 2Department of Risk, Safety, and Security, Institute for Energy Technology, 1777 Halden, Norway; fabien.sechi@ife.no; 3Department of Control Room and Interaction Design, Institute for Energy Technology, 1777 Halden, Norway; hanne.rodsethol@ife.no

**Keywords:** social robots, human–robot interaction, security, public space, cybersecurity, privacy, safety

## Abstract

Background: As social robots increasingly integrate into public spaces, comprehending their security implications becomes paramount. This study is conducted amidst the growing use of social robots in public spaces (SRPS), emphasising the necessity for tailored security standards for these unique robotic systems. Methods: In this systematic mapping study (SMS), we meticulously review and analyse existing literature from the Web of Science database, following guidelines by Petersen et al. We employ a structured approach to categorise and synthesise literature on SRPS security aspects, including physical safety, data privacy, cybersecurity, and legal/ethical considerations. Results: Our analysis reveals a significant gap in existing safety standards, originally designed for industrial robots, that need to be revised for SRPS. We propose a thematic framework consolidating essential security guidelines for SRPS, substantiated by evidence from a considerable percentage of the primary studies analysed. Conclusions: The study underscores the urgent need for comprehensive, bespoke security standards and frameworks for SRPS. These standards ensure that SRPS operate securely and ethically, respecting individual rights and public safety, while fostering seamless integration into diverse human-centric environments. This work is poised to enhance public trust and acceptance of these robots, offering significant value to developers, policymakers, and the general public.

## 1. Introduction

The world has witnessed a significant surge in deploying social robots in public spaces (SRPS) in recent years [1]. These SRPS are designed to collaborate with humans and have found applications across diverse environments, including retail [2], healthcare [3], education [4], and public services [5]. As SRPS function autonomously in environments characterised by frequent and complex human interaction, security issues surrounding their deployment are becoming increasingly critical.

Social robots (SRs) are autonomous, physically embodied agents that interact and communicate with humans or other autonomous agents based on social behaviours and predefined rules using their sensors and actuators [6]. They represent a transformative movement in robotics that has the potential for profound societal impacts, pending their attainment of adequate levels of autonomy, AI capability, safety, and security for widespread utilisation [7].

However, integrating robots into public and private domains introduces the challenge of assuring the security and safety of human interactions [8]. The complexities of public spaces, with their intricate and unpredictable nature, raise significant stakes involving various stakeholders and interaction dynamics [9]. SR could potentially amass and analyse vast amounts of personal information, akin to social media platforms and digital smartphone assistants. Notably, these robots present a unique challenge: users may have limited control over data collection processes, creating a potential imbalance between technology and individual privacy rights [10].

The rising prominence of SRPS accentuates the need for an in-depth, systematic exploration of their security facets. There is an urgent call to craft robust security standards that cater specifically to these robots, setting them apart from industrial variants. This study addresses this void by offering an exhaustive analysis of SRPS security dimensions. Numerous studies have underscored the importance of safety, security, privacy, and ethical considerations in SRPS [11,12]. While some have delved into the threat landscape and attack vectors [5,8], others concentrated on robotic cybersecurity [13]. Yet, comprehensive research covering all security aspects of SRPS remains sparse.

This systematic mapping study (SMS) is designed to review and analyse the existing body of literature related to the security of SRPS. It seeks to identify and evaluate the current security measures, detect existing gaps and issues, and propose a set of comprehensive guidelines that could serve as a foundation for future security standards for SRPS. In doing so, this study aims to provide valuable insights and direction for researchers, policymakers, robot developers, and the broader community.

The main contributions of this study include the following:Comprehensive literature analysis: This study provides a thorough and systematic review and analysis of the existing literature on the security aspects of SRPS, examining academic papers, reports, and standards across multiple databases. This forms a robust and detailed map of the current knowledge base.Identification of key security themes: The study identifies and categorises the key security themes pertinent to SRPS, including physical safety and integrity, data privacy and confidentiality, communication security, ethical considerations, and others. This thematic framework helps to structure the complex and diverse security issues associated with SRPS.Assessment of existing standards: The study conducts a detailed assessment of existing safety and security standards, revealing gaps where these standards—often developed for industrial robots—are inadequate for SRPS due to their unique operational environments and challenges.Proposal for new security guidelines: Based on the literature analysis and existing standards, this study proposes a set of comprehensive security guidelines specifically tailored to SRPS. These guidelines are designed to be actionable and can be used as a baseline for developing formal security standards for SRPS.Insights into cultural and ethical implications: The study sheds light on the broader cultural and ethical implications of deploying social robots in public spaces, fostering awareness of the need for SRPS to respect human rights, privacy, and social norms.Highlighting future research directions: This study outlines several future research paths for advancing the field, such as exploring cultural differences in SRPS acceptance, developing standardised testing protocols for these robots, and analysing real-world SRPS deployments.Value to stakeholders: The study offers invaluable insights for various stakeholders, including policymakers, roboticists, industry professionals, and the general public. The study provides a foundational resource for policymakers and industry professionals to inform the development of regulations and standards. For roboticists and developers, it offers a clear framework for designing and deploying SRPS with security at the forefront. For the general public, it aims to raise awareness of the potential risks and benefits associated with SRPS.

The remainder of this paper encompasses a review of the pertinent literature on the security of SRPS (Section 2), an account of the study’s methodological approach, including the literature search strategy and analysis framework (Section 3), a detailed presentation and discussion of the findings segmented by key themes and their implications (Section 4), and a concluding overview accentuating the study’s primary insights while proposing directions for future research in this domain (Section 6).

## 2. Related Works

As SRs increasingly become integrated into public spaces, a growing body of literature delves into the various aspects of their deployment. This section reviews relevant works in security, safety, privacy, ethics, and regulatory standards associated with SRPS.

### 2.1. Cybersecurity and Safety of Robots

Numerous studies have been conducted to address the safety and security of robots. For instance, Mavrogiannis et al. [14] explored safety mechanisms to prevent robot collisions with humans in crowded places, while Mayoral-Vilches [13] studied the potential security vulnerabilities in robot software. Unlike our systematic mapping study, these works often focus on industrial robots and lack a comprehensive consideration of SRPS.

### 2.2. Data Privacy Concerns

Significant work has been conducted on the privacy implications of robots in public and private spaces, such as Lutz et al. [15], highlighting potential issues associated with personal data collection by robots. Our study further extends this analysis, identifying specific privacy themes pertinent to SRPS.

### 2.3. Ethical Considerations

Several scholars, such as Boada et al. [11], have delved into the ethical implications of SR, focusing on issues like human–robot interaction ethics and potential emotional manipulation by robots. We build upon this work, contextualising it within the public space setting.

### 2.4. Legal and Regulatory Frameworks

A few studies, like those by Andraško et al. [16], have investigated the legal frameworks surrounding robot deployment. However, these studies often deal with robots in a broader context, rather than SRPS specifically. Our study addresses this gap, offering insights into the unique legal considerations for SRPS.

### 2.5. Human Interaction and Social Acceptance

Research, such as that by Baisch et al. [17], has examined how humans interact with robots and their social acceptance. Our mapping study extends this line of work, focusing on these interactions’ safety and security implications in public spaces.

### 2.6. Cultural Aspects of SRPS

Studies, such as those conducted by Recchiuto and Sgorbissa [18], explored cultural differences in accepting robots in public spaces. We recognise this dimension in our study and propose it as a key direction for future research.

### 2.7. Existing Systematic Reviews

While there are existing systematic reviews related to robotics, such as Vulpe et al. [19], they do not explicitly focus on the security aspects of SRPS. Our study is distinct in its systematic and focused approach to the security aspects of SRPS.

### 2.8. Unique Contribution of This Study

Our systematic mapping study stands out in its comprehensive and structured approach to synthesising the diverse security aspects associated with social robots in public spaces. By meticulously analysing existing literature and standards, we construct a robust thematic framework that consolidates the essential security guidelines for SRPS and highlights current standards and practices gaps.

This review paper significantly enhances the existing literature on SRPS by offering a synthesised and consolidated view of the prevailing research in the field, drawing from a wide array of individual studies to offer a comprehensive perspective that is greater than the sum of its parts.

Firstly, the systematic mapping study delineates and highlights the critical areas of SRPS, including safety standards, data privacy, ethical considerations, and human-centric interaction protocols, fostering a deeper understanding of the multifaceted security landscape surrounding SRPS. It pinpoints the areas where prevailing standards—primarily designed for industrial robots—fall short in addressing the unique challenges posed by social robots in public spaces.

Secondly, it sheds light on the emerging trends and notable developments in the field, presenting a chronological view of the research trajectory, which individual studies might not provide. This gives readers an understanding of the evolutionary path of research in this domain, helping identify both the peaks of heightened research activity and periods of relative stagnation.

Moreover, it spotlights pivotal works and benchmark studies in the sector—like Bryson et al.’s heavily cited paper—helping researchers quickly identify the cornerstone literature in the field and understand the pivotal discourses that have shaped the current understanding of SRPS security aspects.

Additionally, mapping out the thematic areas prominently covered in primary studies offers a snapshot of the focal points of contemporary research, presenting a cohesive picture of the prevailing academic discourse and allowing readers to grasp the central themes dominating the SRPS narrative quickly.

Lastly, it lays out a roadmap for future research, identifying pressing questions that remain unanswered and proposing potential directions for further exploration, thereby acting as a compass that guides forthcoming research to address gaps and venture into unexplored territories.

Thus, the review paper acts as a pivotal resource, weaving individual research threads into a rich tapestry that offers readers an in-depth understanding of the state of SRPS security research, guiding scholars and practitioners to navigate the complex landscape with an informed, comprehensive perspective. It brings crucial insights and perspectives that are not readily apparent when individual papers are viewed in isolation, fostering a nuanced understanding grounded in a rich, multifaceted view of the existing literature.

## 3. Methodology

The methodology for this systematic mapping study (SMS) on “security aspects of srps” is predicated upon the guidelines stipulated by Petersen et al. [20,21] for conducting an SMS, supplemented by Kitchenham’s [22,23] guidelines for a systematic literature review (SLR) and insights from Weidt and Rodrigo [24]. Recognising the overlaps between SMS and SLR, these guidelines are robust foundations for the study.

As illustrated in Figure 1, our proposed methodology unfolds across three stages: planning, conducting, and reporting the SMS. Each stage comprises four key activities, carefully calibrated to ensure a comprehensive and valid exploration of the subject matter.

### 3.1. Planning Phase

The planning phase of this study is designed around four critical components: (i) recognising the need for the SMS, (ii) identifying primary studies, (iii) establishing a method for data extraction, and (iv) identifying and devising ways to mitigate potential threats to validity.

#### 3.1.1. Rationale and Objectives of the SMS

The initial phase involves delineating the purpose and framing the research questions that will steer this SMS. This study was conceived in response to the growing prevalence and complexity of security issues linked to using SRPS. Despite the increasing attention in this area, comprehensive understanding and collation of these aspects remain scattered and fragmented, which creates a clear and urgent need for an overarching, systematic investigation—hence the present SMS.

The principal objective of this SMS is to identify, examine, and categorise the variety of reported security aspects related to the deployment of SRPS. This would serve as a comprehensive foundation for both future academic studies and practical applications, consolidating existing knowledge and highlighting areas that may require further investigation.

##### Research Questions

To effectively address the above objectives, two groups of research questions (RQ) were meticulously formulated:

Group 1: Unfolding Research Trends and Methodologies Concerning Security Aspects of SRPS

RQ 1.1 How has the research focus on the security aspects of SRPS evolved over time? This question aims to provide a historical overview and trace the development and shifts in focus, offering insights into the trajectory of the research field.RQ 1.2 How can insights into the influence and impact of these studies be extracted from their citation network? Understanding the citation network can help identify key studies that have shaped the field, providing an understanding of their relevance and impact.RQ 1.3 What methodologies, types of studies, and thematic areas predominantly characterise research on the security aspects of SRPS? By identifying the methodologies and types of studies used, this question seeks to understand the approaches that have been most effective and prevalent in studying this topic.

Group 2: In-depth Examination of Security Aspects, Normative Guidelines, and Design Principles in SRPS

RQ 2.1 What are the specific security aspects consistently highlighted in studies on SRPS, and how are they defined? This question seeks to identify and understand the primary security concerns linked to SRPS.RQ 2.2 Which guidelines are frequently reported for bolstering the security of SRPS, and what key themes do they encompass? Understanding the existing guidelines for enhancing security can aid in establishing best practices and identifying gaps where new guidelines might be needed.RQ 2.3 What design principles are proposed for augmenting the security of SRPS, and what contributions do they make to the field? Identifying and understanding design principles can provide practical guidance for developing more secure SRPS.

#### 3.1.2. Identification of Primary Studies

The second stage of our process involves establishing the means by which we will source information. This includes the formulation of a search methodology, the development of inclusion and exclusion criteria, and the establishment of quality assessment measures. Each component is integral to ensuring the relevance and reliability of the studies contributing to our SMS.

##### Information Source (Digital Database)

Our search strategy is designed around the use of automatic search techniques via the Web of Science (WoS) Core Collection (“Clarivate” 2022). This comprehensive database provides access to an expansive collection of high-quality scholarly journals, conference proceedings, and other sources of information from reputable publishers such as *ACM, Elsevier, Wiley, Springer, Sage, Taylor & Francis*, and *MIT Press*. The immense size of this collection, containing over 72 million article records and a billion cited references, ensures a rich and varied sample from which we can select the primary studies.

##### Search String Construction

To optimise the efficiency of our search, we underwent several iterations in formulating our search string, focusing on the keywords ‘social robots’ and ‘security’. We utilised wildcards (“*”) to account for plural and inflected forms of words, constructing three search strings. The third search string, which returns results involving both social robots and security, was selected as our final search tool.

##### Filtering Strategy

In order to further refine our search results, we incorporated a two-stage filtering process as suggested by Petersen et al. [20,21] and Kitchenham and Brereton [22,23]. The first stage involved an examination of the titles and abstracts of potential studies. The second stage required a full-text reading of these studies. To expedite this process, when the title and abstract provided sufficient information to reach a decision, only the introduction and conclusion of a given study were consulted.

##### Inclusion and Exclusion Criteria

Developing clear inclusion and exclusion criteria is integral to our search strategy. The following criteria, formulated during the planning phase, guided our selection of studies:

The inclusion criteria encompass the following:Field of Study (IC1): Our study focuses on social robots operating in public spaces, with a specific interest in their security aspects. The definition of “public spaces” encompasses both indoor and outdoor areas that are publicly accessible.Methodology (IC2): We consider studies utilising various research methods, as diverse methodologies can provide a broader understanding of the field.Publication type (IC3): We only include peer-reviewed studies to ensure the credibility of the information used in our SMS.Language (IC4): To standardise the analysis process and eliminate language-related biases, we only include studies published in English.Publication period (IC5): Given the rapidly evolving nature of the field, we focus on studies published between 2016 and 2022 to ensure relevance and recency.

Our exclusion criteria include the following:Out of scope (EC1): We exclude studies that do not align with our specific domain of interest. For example, we omit studies concentrating on the appearance, acceptance, trust, and application of social robots unless they specifically address security aspects.Secondary studies (EC2): Secondary studies are excluded to ensure that our SMS is built upon primary research.Non-English language (EC3): Non-English studies are excluded to eliminate any potential inaccuracies stemming from translation.Duplicate studies (EC4): Duplicated studies or extended versions of original papers are excluded to avoid redundancies.Inaccessible studies (EC5): To maintain transparency and reproducibility, any studies that are not openly accessible are excluded.Front and back matter (EC6): Any search results that only contain front or back matter are excluded, as they do not contribute meaningful data.Non-peer-reviewed papers (EC7): Studies that have not undergone the peer-review process are excluded to ensure that only quality research contributes to our SMS.

##### Quality Assessment

The WoS core collection is a trusted source of high-quality research. In addition to this, our methodology incorporates a snowballing strategy, checking the references of included studies to locate further relevant work. This helps to ensure a comprehensive representation of the field. The introduction of several exclusion criteria ensures that only publications of good quality are included in our study. While Petersen, Kitchenham, and Weidt and Silva emphasise that the quality assessment is not a priority for SMS, we take this step to further ensure the reliability of our results, as our study intends to provide not just an overview but a comprehensive, high-quality mapping of the emerging research area.

#### 3.1.3. Data Extraction

The third step involves devising a data extraction strategy and form. This measure assures that all relevant information is systematically gathered and organised, thus enhancing the robustness and comprehensibility of the analysis. Three data sets were extracted: the entire search results from our WoS search, from which two filtering stages (title/abstract and full-text) were conducted, the meta-data of all included studies, and the data answering our research questions. Extraction was performed using the Zotero reference manager, version 6.0.26, for the first two stages.

Data on research trends and security aspects were targeted for the final stage. Research trend data included study ID, title, publication year, author(s), their affiliations, study source, type and focus, and the research method employed. Security aspect data encapsulated reported security aspects and principles. A detailed data extraction form is provided in Appendix B, and all extracted data are available in our Appendix A [25].

#### 3.1.4. Validity Threats Identification and Mitigation

The final step of the planning stage focuses on recognising and mitigating potential threats to the validity of this SMS. This analysis ensures the reliability of our findings. Below, we identified key threats and their corresponding mitigations using insights from Ampatzoglou et al. [26], and Sjøberg and Bergersen [27].

##### Selection Bias

Selection bias may occur during the identification of primary studies due to the inclusion/exclusion criteria or the chosen databases.

Mitigation: We used well-defined inclusion/exclusion criteria and a comprehensive search string to minimise selection bias. Further, the Web of Science (WoS) Core Collection, a broad and globally recognised database, was employed to ensure the wide coverage of literature.

##### Search String Limitations

The search string might not encompass all relevant studies or may bring in unrelated studies.

Mitigation: The search string was iteratively tested and refined to ensure that it is precise and wide reaching. Wildcards were used to incorporate variations of the keywords.

##### Data Extraction Errors

Data extraction can introduce errors or inconsistencies if not performed systematically and accurately.

Mitigation: A detailed and systematic data extraction form was used. Two researchers also performed the extraction process independently to reduce errors and bias.

##### Interpretation Bias

There might be a bias in interpreting the results of the extracted data, leading to skewed findings.

Mitigation: Findings were cross checked among the research team members to avoid individual interpretation bias. Disagreements were resolved through consensus or by involving a third researcher.

##### Quality Assessment

There is a risk that the quality of primary studies may affect the quality of the SMS findings.

Mitigation: We included only peer-reviewed studies and applied specific exclusion criteria to eliminate non-peer-reviewed papers. Snowballing was performed to increase the sample representation and ensure a good-quality SMS.

By pre-emptively considering these threats and their mitigations, we aim to ensure the validity and reliability of the study’s findings.

##### Addressing Search String Specificity and Incorporating Varied Terminologies

In constructing our research methodology, it was pertinent to formulate a search string that adequately and precisely captures the wealth of scholarly works central to the security aspects of SRPS. Acknowledging the diverse terminologies pervading this vibrant field—ranging from “robot companions” and “humanoid robots” to more specific use cases like “office robots”—we endeavoured to engage in an iterative process of testing and refining our search string. Initial explorations incorporated broader terminologies, such as “human–humanoid interaction (HHI) systems” and “artificial social intelligence”. However, we found that a wider net fetched many unrelated studies, steering away from the focused trajectory of SRPS security that our research mandated. Thus, a judicious choice was made to optimize the search string to hone in on works most aligned with our core investigation, settling on a wildcard approach with “social robot*” to maintain a precise yet inclusive search periphery. While this strategy equipped us with a rich research repository to base our study on, we concede the potential exclusion of relevant works utilising alternative lexicons. In cognizance of this, we advocate for subsequent reviews to embrace an expansive approach, welcoming the dynamism and evolving lexicon within this interdisciplinary realm. Future endeavours in this domain would benefit from a panoramic view, thus fostering a richer, more nuanced understanding of SRPS security dynamics.

### 3.2. Conducting Phase

The execution of this study includes four core steps: (i) searching the WoS Core Collection, (ii) filtering the search results, (iii) assessing the quality of the filtered results, which leads to the final selection of studies, and (iv) snowballing the reference sections of all included studies to identify additional relevant works for this SMS.

#### 3.2.1. Searching the WoS Core Collection

We initiated our search for pertinent primary studies on 23 July 2023, deploying three distinct search strings. The first string was designed to identify studies related to social robots, resulting in 1874 studies. As this was an automated search, we implemented suitable inclusion and exclusion criteria from the outset, such as “all fields”, publication years (2016–2022), article types (articles, early access, proceeding papers, and book chapters), and language (English). The second search string was tailored towards security-related papers and yielded 174,251 results. Finally, the third search string, focusing on the intersection of the two domains, identified 30 studies. These 30 studies formed the initial data set for the filtering stage.

We exported the resulting 30 studies from the WoS platform in Bibtex format and imported them into the Zotero reference manager. Utilising a reference manager ensures uniformity and consistency, and helps prevent errors during meta-data extraction. From Zotero, the 30 files were exported in CSV format to Microsoft Excel for further filtering (please refer to sheet 1 titled “search_result” in our Appendix A for details).

In order to promote the reproducibility of our study, we document the search strings, the results they produced, and their corresponding URL links in Table 1.

#### 3.2.2. Refinement of Search Results and Quality Assessment

Our study utilised a rigorous, two-stage filtration process for the search results. Initially, we applied the specific inclusion and exclusion criteria to the titles and abstracts of the 30 identified studies. These details are in our Appendix A [25]. As a result of these criteria, 11 studies were deemed “EC1: Out of Scope”. Among these, seven did not have SRPS research as their primary focus, while the other four failed to address the security aspects of SRPS sufficiently. Consequently, 19 studies were earmarked for thorough full-text scrutiny.

During the second filtration stage, an additional five studies were eliminated. Although SRPS and security were mentioned in the abstracts or titles of four of these studies, they failed to address our research questions concerning security aspects satisfactorily. The final study was eliminated due to language constraints, as it was unavailable in English.

Upon completion of the filtration process, the remaining 14 studies underwent a comprehensive quality assessment. The findings from this evaluation confirmed their high quality and appropriateness for addressing the research questions of this study. Each of the 14 studies was assigned a unique Study ID (ranging from P01 to P14) for ease of identification. The details of these studies were then exported in CSV format to a Microsoft Excel worksheet (titled “Primary Studies” under worksheet 2 in our Appendix A) [25].

#### 3.2.3. Backward Snowballing

We undertook a backwards snowballing procedure to align with the guidelines underpinning this study. This process involved reviewing the references of the 14 included primary studies to unearth further relevant studies that could contribute to this SMS. Given that the purpose of this SMS is to compile a summary map of security aspects pertaining to SRPS, rather than conducting a systematic literature review (SLR) encompassing all evidence in the research domain, the snowballing process was completed in a single round.

During this snowballing phase, a total of 998 references were scrutinised against the study’s inclusion and exclusion criteria. This evaluation was conducted initially using titles and abstracts in round one, with 69 of these references advancing to full-text assessment. Their potential to address our research questions and fulfil our study’s quality criteria was then evaluated.

After the second round, we pinpointed 26 studies that met the inclusion and exclusion criteria, making them eligible for our study’s inclusion. Snowballing from 8 primary studies (P03, P04, P05, P06, P10, P11, P13, and P14) yielded these 26 relevant studies. To bolster our study’s reproducibility, we documented each step of the backward snowballing process. This includes detailing the number of references per study and the associated study IDs of the incorporated studies, all of which are comprehensively presented in Table 2. For example, from the snowballing process, P03 generated three papers, labelled as P03S1, P03S2, and P03S3, facilitating any reader’s efforts to retrace and replicate our methodology.

### 3.3. Reporting Phase

The concluding phase of this study revolves around preparing and presenting the primary studies for comprehensive reporting. This phase encompasses four interconnected activities: showcasing the 40 primary studies, extracting pertinent data from these studies to address our research questions, categorising and analysing the extracted data, and creating security aspects maps for SRPS. The subsequent section, “Results and Discussion”, furnishes a thorough summary of the reporting process and findings of this study.

## 4. Results

This section presents the results of our study in the context of our initial research questions. Of the 176,281 initial studies, only 14 directly addressed SRPS and security. The backward snowballing of these studies’ references led to 26 additional papers, culminating in 40 core studies for our systematic mapping.

### 4.1. Group 1: Unfolding Research Trends and Methodologies Concerning Security Aspects of SRPS

#### 4.1.1. How Has the Research Focus on the Security Aspects of SRPS Evolved over Time

In addressing the evolution of research focus on security aspects of SRPS, we analysed the temporal distribution of our primary studies, considering their publication year, study type, and the security topics of focus.

From our analysis, the composition of the primary studies indicates a predominant preference for journal articles, making up 25 of the total, followed by 12 conference papers, two book sections, and a single book. This suggests that most research on this topic is published in peer-reviewed journals, indicating the academic significance and rigour of the subject matter. The summary of these studies is presented in Table 3, which details the 40 primary studies, including their unique study identifiers, authors, titles, and citation counts.

Regarding the source of these studies, most contributions originate from *SpringerLink*, with 13 papers. This is followed by *IEEE Xplore* with nine papers, while *ACM* and *MDPI* each contributed four. Other notable contributors include *ScienceDirect* and *Science Robotics*, with three and two papers, respectively, while *Taylor & Francis, Emerald, IOActive, JMIR*, and *AJIS* each have one. This wide range of sources illustrates the cross-disciplinary interest in SRPS security aspects, extending beyond the traditional computer science and engineering fields.

Looking at the publication years of the primary studies, we see a relatively steady increase from two papers in 2016, peaking at nine papers in 2017, and then a more gradual rise to eight papers in 2019. The subsequent years (2020 and 2021) both contributed six papers, with a slight decrease to five papers in 2022. This pattern may suggest an increasing interest and recognition of the importance of SRPS security aspects within the research community, with 2017 marking a notable spike in publications. The relatively consistent numbers from 2019 to 2022 suggest that this field of study maintains its relevance, ensuring a continuous stream of research despite the emergence of COVID-19 restrictions that affected all research activities. This may also account for the slight decrease in 2022. A comprehensive summary of the visual representation of the temporal distribution of these papers is shown in Figure 2.

This in-depth analysis not only answers our research question regarding the evolution of focus on the security aspects of SRPS but also presents key insights into the state and progress of the research field.

To further address the research question, “How can insights into the influence and impact of these studies be extracted from their citation network?” it is essential to consider the authors’ affiliations and their respective backgrounds, either from academia, industry, or both. Table 4 shows that most of the studies originate from academic institutions. This underscores the fact that foundational research, theories, and principles in the domain of SRPS security largely arise from the academic sphere. Such research often lays the groundwork, offering insights, methodologies, and frameworks upon which practical applications can be built.

While academic contributions dominate, including industry-driven research (like P03S2), which emphasises real-world applications and pragmatic solutions, often addresses immediate challenges faced in practical implementations, offering solutions that may be directly applicable to existing systems or technologies. Studies with a mixed affiliation, combining both academia and industry (such as P09, P05S8, P10S17, P13S22, P13S23, and P13S24), are particularly noteworthy. These collaborations bridge the gap between theoretical research and practical application, suggesting a comprehensive approach that addresses both foundational challenges and real-world problems. Collaborative efforts can yield research that benefits from the depth of academic investigation and the hands-on experience of industry professionals. Considering the rapid technological advancements in SRPS security, fostering collaborations between academia and industry becomes imperative. While academic studies delve deep into theoretical constructs, real-world testing and validation are often the domain of the industry. Combining these strengths can lead to more robust, validated, and universally applicable solutions. In addition, collaborations can ensure that research is both forward thinking (from the academic side) and grounded in real-world challenges (from the industry side). Such a balanced approach can lead to revolutionary and practical innovations.

In conclusion, while analysing the citation network to determine the influence and impact of various studies, one should also consider the authors’ affiliations. The blend of academia, industry, and collaborative efforts provides a multi-faceted perspective on SRPS security. Encouraging more collaborative efforts between the two domains will likely enrich the field, leading to deeply insightful and practically relevant insights.

#### 4.1.2. How Can Insights into the Influence and Impact of These Studies Be Extracted from Their Citation Network?

To answer the research question at hand, it is crucial to examine the citation patterns of the primary studies listed in Table 5. This analysis will illuminate the influence and impact of these studies in the field of socially relevant public space (SRPS)security. We can gauge this influence by looking at both the total number of citations each paper has received and their average annual citations on Google Scholar on 3 August 2023, the latter taking into account the duration a paper has been published and available to be cited.

Bryson et al.’s “Of, for, and by the people: the legal lacuna of synthetic persons” (P06S9) stands prominently at the pinnacle of our citation list. Released in 2017, this study has garnered 292 citations, averaging an impressive 49 citations annually. This substantial citation frequency underscores the paper’s profound impact on the discipline, implying that its legal perspectives have been instrumental in subsequent research. This further emphasises that the legal dimensions of SRPS remain a crucial focal point in discussions surrounding SRPS security.

Another significant paper is “Transparent, explainable, and Accountable AI for Robotics” by Wachter et al. (P03S3) from 2017. This study boasts 237 total citations and an average annual citation rate of 40. The paper’s emphasis on transparency and accountability in AI for robotics appears to have resonated broadly, underscoring its importance in contemporary discussions around SRPS security.

In our set of studies, Eduard Fosch-Villaronga emerges as a recurrent author, having authored or co-authored five papers (P06, P06S12, P06S13, P06S14, and P10S16). Despite the individual papers having a lower total citation count (ranging from 21 to 57), Fosch-Villaronga’s collective contribution to the field is significant. Their focus on the legal and ethical aspects of robotics signals these issues’ prominence within SRPS security research.

The average annual citation rate offers insight into the relative impact of more recent articles. For instance, “We need to talk about deception in social robotics!” by Sharkey and Sharkey (P03S1), published in 2021, has an annual citation average of 37, suggesting a potent initial impact despite its recent publication.

In conclusion, the citation network’s analysis provides valuable insights into the influential studies, dominant themes, and leading authors within the SRPS security field. However, it is important to remember that citation count is just one measure of influence and impact, and other factors, such as the quality of the research and its applicability to real-world scenarios, are equally important. A visual representation of the research trends in terms of citation is presented in Figure 3.

#### 4.1.3. What Methodologies, Types of Studies, and Thematic Areas Predominantly Characterise Research on the Security Aspects of SRPS?

To address the research mentioned above question, we delved into the research types, methods, and thematic areas outlined in Table 6. Our analysis categorised research types into solution proposals, philosophical/conceptual research, validation research, experience reports, and focus groups, in line with the study guidelines we followed. It is worth highlighting that several papers employed multiple research approaches. We distinguished research methods between quantitative, qualitative, and mixed methodologies. We further classified the thematic areas into five broad categories: technical security solutions, ethical and philosophical implications, legal and regulatory concerns, interaction and behaviour, and privacy concerns. For a more granular thematic breakdown comprising 13 categories from our primary studies, please refer to Table 6.

##### Research Types

Our SMS unveiled the following classifications of studies:**Solution proposals:** These articles chiefly aim to identify challenges and propose technical or methodological remedies. It is the predominant research type in our data set with 18 studies (P01, P03, P04, P05, P07, P12, P13, P03S2, P05S6, P05S7, P05S8, P06S10, P06S14, P11S21, P13S22, P13S23, P13S24, and P14S25).**Philosophical/conceptual analyses:** These engage with the philosophical dimensions, ethical nuances, or critical discussions related to the topic. Thirteen studies (P02, P06, P10, P14, P03S1, P03S3, P06S9, P06S12, P06S13, P06S14, P10S16, P10S18, and P14S26) belong to this realm.**Evaluation research/reports:** These appraise specific facets or occurrences. There are seven studies (P09, P11, P03S2, P04S5, P06S15, P11S20, and P13S23) in this classification.**Validation studies:** These are dedicated to endorsing particular hypotheses or systems. Papers P08, P06S15, and P14S25 are representatives of this category.**Experience reports:** These elucidate findings and lessons drawn from specific experiences or enactments. Studies P06S11, P10S17, and P10S19 exemplify this category.**Focus groups:** We identified a singular focus group, labelled as P04S4.

##### Research Methods

Through our SMS, we discerned the following methodologies utilised in the studies:*Qualitative:* These studies gravitate towards non-numeric data, leveraging observations, discussions, or narrative interpretations. A total of 15 papers (P02, P06, P10, P03S1, P03S3, P04S4, P04S5, P06S9, P06S11, P06S12, P06S13, P10S16, P10S18, P10S19, and P14S26) in our compilation adopted this approach.*Quantitative:* Here, the emphasis is on numeric data, frequently accompanied by statistical scrutiny. In our data set, ten studies (P07, P08, P11, P13, P14, P03S2, P05S7, P06S15, P11S21, and P14S25) predominantly employed this method.*Mixed:* A significant portion of the studies, numbering 15 (P01, P03, P04, P05, P09, P12, P03S2, P05S6, P05S8, P06S10, P06S14, P06S15, P10S17, P13S23, and P13S24), amalgamated both qualitative and quantitative methods for a comprehensive analysis.

##### Thematic Insights

From our SMS, we derived seven core thematic areas. They are as follows:**Cybersecurity**Encompasses a broad range of topics, from network, application, and cloud security to user education, identity management, and cybersecurity regulations.Included papers: P03, P04, P05, P06, P07, P10, P13, P03S2, P06S10, P06S13, P10S19, P13S22, P13S23, and P13S24.Represents 35% of our primary studies.**Safety**Discusses structural safety, motion, fail-safe mechanisms, protection against cyber–physical threats like stalking, emergency responses, and safety during maintenance.Included papers: P01, P11S20, and P11S21.Represents 7.5% of our primary studies.**Privacy**Delves into data collection, consent, anonymisation, transparency, and behavioural privacy. Topics like data security and control have been grouped under cybersecurity.Included papers: P03, P04, P04S4, P04S5, P06S13, and P14S26.Represents 15% of our primary studies.**Reliability and continuity**Covers aspects such as hardware and software reliability, maintenance, network connectivity, fault tolerance, and user interface consistency.Included papers: P11, P13, and P05S6.Represents 7.5% of our primary studies.**Legal challenges**Focuses on data privacy laws, cybersecurity standards, SR liability and insurance, telecommunication regulations compliance, and public space-specific regulations.Included papers: P10, P11, P13, P05S6, P06S12, P06S14, and P06S16.Represents 17.5% of our primary studies.**Ethical concerns**Encompasses themes like human autonomy, informed consent, justice, transparency, and sustainability.Included papers: P02, P06, P10, P03S1, P03S3, P06S11, P06S15, and P10S18.Represents 22.5% of our primary studies.**Influence and manipulation:**Explores user profiling, SR’s persuasive capabilities, and accountability mechanisms.Includes papers: P08, P09, P11, P12, P05S8, P14S25.Represents 15% of our primary studies.

Notably, many studies span multiple themes. We categorised each based on its dominant focus, although some touch upon areas like cybersecurity, data privacy, and legal aspects, making them relevant for various categories. An emergent theme also noted is social awareness and campaigns related to SRPS, including studies like P02, P08, P09, P03S1, P03S3, P06S9, and P14S25. It is worth highlighting that the dominant thematic areas represented in our primary studies are cybersecurity (35%), ethical concerns (22.5%), legal frameworks (17.5%), data privacy (15%), and user influence and manipulation (15%).

Research on the security aspects of SRPS is diverse and multifaceted, spanning from highly technical solution proposals to philosophical debates. The predominant methodologies range from mixed methods to qualitative and quantitative studies. At the same time, the thematic areas broadly encapsulate technical security solutions, ethical implications, legal and regulatory concerns, human–robot interaction behaviour, and privacy issues.

### 4.2. Group 2: In-Depth Examination of Security Aspects, Normative Guidelines, and Design Principles in SRPS

#### 4.2.1. Specific Security Aspects in SRPS Studies: Consistent Highlights and Definitions

To delve into this research question, we harnessed our findings from the Group 1 results, exploring the seven distinctive themes to elucidate the security facets of SRPS more clearly. A meticulous analysis is available in our study’s Appendix A, with each claim substantiated by relevant quotations from the Microsoft document titled “Security Aspects Repository” [25]. The security aspects consistently emphasised in our studies, along with their definitions, are as follows:**Cybersecurity of SRs and users:** This aspect primarily concerns safeguarding SRPS systems, networks, and data from cyber threats. Common discussions in the studies pertain to the unique security challenges in SRPS, a broad array of security services ranging from secure bootstrapping and communication to data storage, software updates, and device management, and threats such as unauthorised publishing, unauthorised data access, and denial of service (DoS) attacks. A total of 21 papers, accounting for 52.5% of our primary studies, cover this aspect.**Data privacy of users:** Central to this domain is protecting user data, ensuring data handling that respects individual rights. Major themes involve communication security risk assessments, the enactment of access control policies, implications of AI and robots on privacy, championing the principle of “Privacy by design”, and emphasising that domestic robots should always respect user privacy expectations. This theme is covered in 11 papers, making up 27.5% of our primary studies.**Physical safety of SRs and users:** The focus here is on ensuring the safety of both users and the physical structure of the robots. Studies underscore the significance of safety, reliability, and continuity, particularly in socially assistive robots, and the imperative for designing SRPS systems with user safety as a paramount concern. This aspect is detailed in 11 papers, representing 27.5% of our primary studies.**Reliability and continuity of SRs:** This theme emphasises the crucial need for robots to function reliably and consistently. The importance of both reliability and continuity is frequently underscored in contexts such as socially assistive robot scenarios, and the discussions often touch upon the need for industrial control systems to remain operational even in challenging conditions. Six of our studies, which constitute 15% of the primary research, delve into this facet.**Legal framework for SRPS:** This dimension intersects the realms of technology and legal compliance. Key discussions revolve around the necessity of adhering to established legal norms pertinent to SRPS security, safety, and privacy. Additionally, there is a focus on the debate surrounding robot legal personhood and challenges associated with data recording and logging, especially when personal data are in play. Fourteen of our papers, representing 35% of the primary studies, focus on this area.**Ethical Consideration for SRPS:** This domain navigates the intricate waters of moral and philosophical considerations associated with robot deployment. Research consistently brings forth issues such as the inherent challenges in designing machines capable of ethical decision making, the myriad ethical dilemmas users may face when interfacing with AI technology, and the intertwined ethical and legal responsibilities of robot behaviour. This theme finds mention in 16 of our papers, encapsulating 40% of the primary studies.**User influence and manipulation:** This more nuanced theme seeks to understand the potential of robots to subtly shape or alter user behaviours, decisions, and perceptions. Central to this discourse is the unwavering commitment to user security in all its facets, from physical well-being to data integrity. The recurring motif emphasises robots operating reliably, upholding legal mandates, and embedding ethical considerations in both their design and operational paradigms. Five studies, constituting 12.5% of our primary research, touch upon this aspect.

A concise summary of the findings can be found in Table 7.

#### 4.2.2. Key Security Guidelines for Social Robots in Public Spaces

For the safe and ethical deployment of SRPS, it is crucial to ensure they neither compromise personal safety, privacy, and security nor behave unethically. Given the nascent stage of SRPS with multifaceted research areas, security guidelines are a composite of these facets. Our study highlights the following paramount themes for SRPS security guidelines:**Physical safety and integrity:** Robots must have systems to detect and prevent unwanted contacts, operate safely, and feature easily accessible emergency stop functions. This is supported by 27.5% of our primary studies and further buttressed by standards like EN ISO 13482:2014. Safety standards originally aimed at industrial robots should be revamped for the unique demands of SRPS. New safety standards tailored specifically for SRPS are essential, as the current standards do not adequately address the unique safety requirements inherent to SRPS [66].**Data privacy and confidentiality:** Robots should not record, store, or transmit personal data without explicit users’ consent. If data are recorded, strong encryption should be used to protect them. Data collection should adhere to data protection regulations and guidelines such as GDPR and national data protection laws. More than 11 papers in our study reaffirm this theme (P02, P03, P04, P05, P06, P07, P12, P13, P14, P03S1, P03S2, P03S3, P04S4, P04S5, P05S7, P06S12, P06S14, P10S17, P14S25, and P14S26). Applicable privacy standards include NIST SP 800-122 (guide to protecting the confidentiality of personally identifiable information—PII), NIST SP800-53 (security and privacy controls for federal information systems and organisation), ISO/IEC 27701 (privacy information management), and ISO/IEC 27018 (code of practice for protection of PII). The existing data privacy and confidentiality standards, if adequately implemented in SRPS, could address any concern in this aspect.**Communication security:** Communication between robots and control servers or other devices should be encrypted. Secure protocols like TLS should be used for any data transmission. Network vulnerabilities should be regularly assessed and patched. SRPS heavily rely on wireless communication due to their mobile and autonomous nature. Applicable standards include 3GPP 5G advance standards (security architecture and procedures for 5G system release 18—TS 33.501 v 18.0.0); WPA3: Wi-Fi Protected Access III by the Wi-Fi Alliance; NIST SP 800-77, 800-52, and 800-113, addressing different aspects of IPsec, SSL, and TLS security; and IETF (Internet Engineering Task Force) RFC 8446 on TLS v1.3 and 7296 on IKEv2, widely used for IPSec VPNs. Use cases heavily influence communication security standards; hence, there is a need for SRPS use cases and standards [10]. Almost all papers addressing cybersecurity reaffirmed the need for communication security as a theme.**Authentication and authorisation:** Only authorised personnel should have access to the robot’s controls and data. User roles and permissions should be clearly defined. Passwords or other authentication methods should be regularly updated and changed. This theme is a subset of the cybersecurity of SRPS. Applicable security standards reaffirming this theme include (i) NIST SP 800-63 focusing on digital identity guidelines, and (ii) IETF RFC 6749 (OAuth 2.0) and RFC 4120 (The Kerberos network authentication service v.5), among others.**Operational transparency:** The robot should clearly indicate when it is recording or collecting data. Robots should be easily identifiable with visible markings or badges. This theme is a subset of data privacy. Applicable standards include IEEE P7001-2021 [67] focusing on measurable, testable levels of transparency for autonomous systems [68], and ISO/TS 15066 [69], focusing on collaborative robot systems and include safety requirements that can be used as a foundation for transparency around safety.**Robustness against cyber attacks:** The robot’s software should be regularly updated to patch known vulnerabilities. There should be a mechanism to detect and respond to any unauthorised intrusion or malware. For our SMS, this theme is a subset of cybersecurity. Applicable standards include (i) ISO/SAE 21434—focusing on cybersecurity risk management for autonomous systems, including threat analysis and vulnerability assessments; (ii) ISO/IEC 27001 ong general information security; (iii) IEC 62443 focusing on industrial network security, which can be applied for broader robotic applications; and (iv) NIST SP 800-183 focusing on IoT security, which can be adapted for SRPS.**Human interaction protocols:** SR should have guidelines on interacting with different age groups, especially vulnerable populations like children, people living with disability, and senior citizens. SR should be designed to understand and respect social norms and boundaries. This guideline is reaffirmed by P09, P12, P03S1, and P03S2, among others of our primary studies. Most existing standards focus on users’ safety, privacy and security. This theme needs special attention for the successful adoption of SRPS.**Monitoring and reporting:** Robots should be monitored for any irregular or unintended behaviours. Any security breaches or unusual events should be logged and reported promptly. Again, this theme is extensively covered by most cybersecurity themes of SRPS.**Environment-aware operation:** The robots should know their environment and adjust their operation mode accordingly. For instance, a robot should operate differently in a crowded space than an empty one.**Regular testing and validation:** The robot’s systems should be regularly tested to ensure that they function correctly and safely. Various scenarios can be simulated to validate the robot’s response to security and safety situations.**Ethical considerations:** Always consider the ethical implications of deploying SR, especially regarding privacy and human rights. Guidelines and policies should be in place to prevent misuse or unethical behaviour by robots. Several calls, debates and concerns about the ethical implications of SRPS need to be standardised.

By diligently following these guidelines, developers and operators of SRPS can proactively establish a secure operational framework that prioritises the protection of individual rights and the safety of the public. A visualisation of the above proposed security guidelines is presented in Figure 4.

#### 4.2.3. What Design Principles Are Proposed for Augmenting the Security of SRPS, and What Contributions Do They Make to the Field?

The research question aims to explore the design principles proposed to enhance the security of service robot and personal service robot systems (SRPS)and evaluate their contributions to the field of robotics security. The findings indicate that 11 out of 40 studies, accounting for 25%, directly addressed security design principles. Numerous security design principles can be inferred from other studies, although they were not explicitly referenced.

The primary security design principles identified for SRPS and their contributions to the field are as follows:**Security by design:** This principle emphasises integrating security measures and considerations into the system’s design and architecture from the outset. It ensures that security requirements, controls, and mechanisms are incorporated throughout the development lifecycle. This creates inherently secure and resilient systems, reducing vulnerabilities and potential threats. Studies P03, P07, P13, and P13S22 explicitly referenced this principle, while others indirectly suggested its application.**Privacy by design:** This principle focuses on proactively integrating privacy considerations into developing systems, products, and processes. It ensures that individuals’ privacy is safeguarded from the start, contributing to enhanced data protection, regulatory compliance, user trust, and ethical innovation. P06S15 and P14S26 explicitly mentioned this principle, while other studies implied its application.**Human-centred design:** This approach places users and their interactions at the core of design processes, creating user-friendly and relevant solutions. It contributes to security by ensuring user-friendly security features, usable authentication methods, accessible security, and promoting user trust in technology adoption. Studies P09, P12, P03S1, P03S2, and P03S3 directly referenced this principle.**Least privilege:** This principle dictates granting users and processes the minimum necessary access rights for their tasks. It limits attack surfaces, unauthorised actions, lateral movement during attacks, and the impact of breaches. Studies P07 and P03S2 explicitly mentioned this principle, while others indirectly referred to it.

Additionally, other security design principles such as transparency, accountability, secure software development, the separation of components, open design, defence in depth, and fail-safe defaults were implied but not explicitly mentioned in the primary studies.

In summary, the contributions of these security design principles to the SRPS field include the creation of secure and resilient systems, protection of user privacy, user-friendly security measures, regulatory compliance, and enhanced user trust. As the SRPS domain continues to evolve, more standardisation and regulations related to security design principles are anticipated to emerge in the future.

## 5. Discussion

### 5.1. Principal Insights and Practical Applications Derived from the Study

Key findings from this SMS include the following:**Emerging security and safety standards** Our study emphasises the pressing need for updated and comprehensive security and safety standards tailored to address the unique requirements of SRPS. Current standards, predominantly based on industrial robots, are ill-suited to govern the new social robots interacting closely with the public in diverse environments.**Data privacy and integrity** As identified in our research, robust data privacy and confidentiality measures are essential. With myriad potential data interactions between SRPS and the public, strict adherence to data protection laws, such as GDPR, and the implementation of strong encryption protocols are imperative.**Human-centric interaction protocols** Our study reiterates the need for clear and ethical interaction protocols, particularly when robots interact with vulnerable populations. These protocols should be informed by a deep understanding of social norms and human behaviour, advocating for respect and empathy in robot design.**Responsive and adaptive operations** SRPS need to be cognizant of their environments and capable of adapting their behaviours accordingly. This ensures the safety of the individuals they interact with and the robots’ integrity.**Ethical considerations** Our study underscores the burgeoning debate on the ethical implications of SRPS. As these robots become more integrated into public life, they must be designed and programmed to respect human rights and operate within clear ethical boundaries.

The practical implications of our findings from this SMS are manifold, deeply influencing several sectors, including robotics design, regulatory frameworks, and deployment strategies. Let us delve into each one:**Robot Design***Addressing safety and ethical considerations through human-centric interaction protocols:* Our findings stress the need for SRPS to have clearly defined ethical interaction protocols, especially when engaging with vulnerable population groups. This implies that robots should be designed to understand and respect human norms and behaviours, ensuring that empathy and respect are central to their programming.*Ensuring data privacy through robust security protocols:* The highlighted need for stringent data privacy and integrity measures necessitates that social robots be equipped with sophisticated security protocols to guard against data breaches and ensure compliance with the GDPR and national applicable laws.**Regulatory Frameworks***Addressing legal aspects through tailored legal frameworks:* The study suggests a significant gap in the existing legal frameworks to accommodate the unique challenges SRPS poses. Legislators can draw insights from this study to craft laws that govern the operation and deployment of SRPS, addressing issues like user influence and potential manipulations.*Addressing ethical concerns through ethical oversights:* Regulatory bodies would benefit from establishing committees to oversee the ethical dimensions of SRPS, ensuring that they operate within defined ethical boundaries and respect human rights. This could also involve formulating standardised testing and validation protocols for evaluating SRPS before deployment.**Deployment Strategies***Employing adaptive operations through responsive SRPS deployment:* When deploying SRPS, it is vital to ensure that they are cognizant of their surroundings and can adapt their behaviours accordingly, safeguarding both the individuals they interact with and maintaining their own integrity.*Improving users’ experience through user-centric design:* Insights from this study should encourage developers to focus on enhancing the user experience, paying attention to aspects like accessibility and the psychological impact SRPS could have on individuals and communities.*Shaping future research through trans-disciplinary collaboration* Stakeholders in the industry should foster collaborations with researchers to delve into the prospective research avenues identified in this study, working towards the secure, ethical, and effective integration of SRPS in public spaces.

In summary, the practical implications of our findings can inform a holistic approach towards the design, deployment, and regulation of SRPS, fostering a landscape where social robots can safely and harmoniously coexist with humans, augmenting public spaces while prioritising security and ethical considerations.

### 5.2. Conceptual Framework Illustrating the Interrelated Key Dimensions of Security in SRPS

Figure 5 outlines a detailed conceptual framework that encapsulates the pivotal security dimensions of SRPS and delineates their intricate interrelationships with pertinent stakeholders. This schematic representation is segmented into four cardinal phases—regulation, production, operation, and interaction—each representing a critical stage in the SRPS lifecycle.

In the regulation phase, the onus rests upon regulators and policymakers to forge comprehensive standards and guidelines grounded in legal, ethical, and governance principles pertinent to SRPS. These directives not only aim to safeguard security and safety but also foster a nurturing environment for the SRPS ecosystem to flourish. Feedback loops are instituted to refine these norms based on real-time insights and developments continually.

The production tier is steered by developers, designers, and manufacturers entrusted with the task of meticulously embedding the legislated standards into the very fabric of SRPS. This involves sculpting security frameworks and instituting privacy safeguards that are both robust and adaptable, thus upholding the precept of security by design. This phase leverages a profound understanding of the comprehensive threat landscape to devise resilient mechanisms against emerging threats.

At the operation echelon, SRPS business proprietors and system administrators come into play, spearheading the efficacious implementation of the prescribed frameworks while advocating for a user-centric approach. This phase is characterised by unyielding vigilance, with regular monitoring and timely updates to mitigate evolving risks and ensure a secure service delivery that stands the test of time.

The interaction phase envisages a collaborative role for the end users, nurturing them to be savvy interactants through the dissemination of awareness and training pertaining to SRPS services. This stage embraces the feedback derived from user experiences, fostering a two-way dialogue to inculcate enhancements that are in sync with user expectations and preferences.

By elucidating the synergistic relationships between these phases and the stakeholders, Figure 5 serves as a beacon, guiding stakeholders at every juncture to foster a secure, ethical, and user-friendly SRPS environment, thereby steering the field towards a future marked by trust and mutual growth.

### 5.3. Delving into the Security Challenges of SRPS: Concrete Instances and Forward-Looking Solutions

Delving deeper into the challenges faced in securing SRPS and addressing them is pivotal in advancing the field. Let us break down these challenges along with potential solutions and their implications:**Data privacy and integrity***Challenges:* The interception of data transferred between SRPS and central servers leading to privacy breaches.*Potential solutions:* Implementing end-to-end encryption and robust authentication mechanisms.*Practical implications:* Enhancing data privacy will build trust among users and foster broader acceptance of SRPS.**Safety standards***Challenges:* The existing safety standards are derived from industrial robot frameworks unsuited for SRPS operating in public spaces with diverse and dynamic environments.*Potential solutions:* Developing safety standards specifically tailored for SRPS, emphasising real-time adaptive safety mechanisms.*Practical implications:* Customised safety standards would ensure the safe interaction of SRPS with humans, mitigating risks and preventing accidents.**Ethical considerations***Challenges:* The potential for SRPS to be used unethically, such as for surveillance or influencing user behaviour subtly.*Potential solutions:* Establishing ethical guidelines that dictate the operations of SRPS, including transparency in data usage and respecting user autonomy.*Practical implications:* Addressing ethical concerns would foster a responsible deployment of SRPS, safeguarding individual and societal values.**Legal frameworks***Challenges:* The current legal frameworks are inadequate to address the unique challenges posed by SRPS, including liability issues in case of malfunctions or accidents.*Potential solutions:* Crafting comprehensive legal frameworks that outline the responsibilities and liabilities associated with deploying SRPS.*Practical implications:* Legal frameworks would provide a clear pathway for accountability, promoting responsible innovation and deployment of SRPS.**Human-centric interaction protocols***Challenges:* Designing SRPS that can appropriately and ethically interact with diverse populations, including vulnerable groups.*Potential solutions:* Incorporating a deep understanding of social norms and human behaviour into the design of interaction protocols, guided by interdisciplinary teams including psychologists, sociologists, and ethicists.*Practical implications:* Human-centric designs would foster positive interactions between humans and SRPS, enhancing the overall user experience and promoting inclusivity.**Cybersecurity***Challenges:* The potential for SRPS to be targeted in cyber-attacks, including hacking and unauthorised control.*Potential solutions:* Developing sophisticated cybersecurity protocols and frameworks, including regular updates and patches to address vulnerabilities.*Practical implications:* Strengthening cybersecurity would protect SRPS from malicious attacks, ensuring their safe and reliable operation.**User influence and manipulation***Challenges:* The possibility for SRPS to influence or manipulate users through persuasive design techniques unduly.*Potential solutions:* Creating guidelines that restrict manipulative design practices and ensure the transparent operation of SRPS.*Practical implications:* Addressing this challenge would preserve user autonomy and prevent potential misuse of SRPS in public spaces.

By addressing these challenges with targeted solutions, we can pave the way for the secure, ethical, and beneficial deployment of SRPS, advancing the field by fostering innovation while prioritising the safety and well-being of the public.

### 5.4. Future Research Directions in SRPS Security

The field of SRPS security is teeming with opportunities for further research. Here are potential directions that can be explored, highlighting areas that necessitate deeper inquiry and the questions that remain unanswered:**Cultural Variability in SRPS Acceptance***Research Area:* Investigating how different cultures perceive and interact with SRPS.*Unanswered Question:* How can SRPS be designed to align with various cultural norms and expectations without compromising security?**Standardised Testing and Validation Protocols***Research Area:* Developing robust and standardised protocols for testing and validating SRPS security measures.*Unanswered Question:* How can these protocols encompass a wide array of scenarios to ensure the readiness of SRPS for deployment in diverse public spaces?**Real-World Case Studies Analysis***Research Area:* Conducting in-depth analyses of real-world SRPS deployments to glean insights into practical challenges and the effectiveness of existing security measures.*Unanswered Question:* What lessons can be drawn from practical deployments to inform more secure and efficient future SRPS implementations?**Comprehensive Security Framework Development***Research Area:* Creating a comprehensive security framework that integrates various aspects like data privacy and ethical considerations into the SRPS development process.*Unanswered Question:* How can this framework serve as a benchmark for evaluating the security readiness of SRPS before their deployment?**Ethical and Regulatory Implications***Research Area:* Exploring the legal landscape and the ethical dimensions governing the deployment and operation of SRPS.*Unanswered Question:* What new laws or amendments are needed to address the unique challenges posed by SRPS, and how can they safeguard individual and community well-being?**User Experience and Human–Robot Interaction***Research Area:* Investigating the psychological impact of SRPS on individuals and communities, focusing on user experience design and accessibility.*Unanswered Question:* How can SRPS be designed to enhance user experience while mitigating potential negative psychological impacts?

By venturing into these areas, future research can build upon the current study’s findings, fostering a secure, ethical, and effective integration of social robots into our public spaces while promoting individual and collective well-being. Each direction provides a pathway to delve deeper into the nuances of SRPS security, addressing unresolved questions and paving the way for groundbreaking advancements in the field.

## 6. Conclusions

As social robots become increasingly prevalent in public spaces, ensuring their secure and ethical operation is paramount. This study systematically analysed the security aspects of SRPS, shedding light on the multi-faceted challenges and the requisite considerations for their deployment. Our findings indicate that the development and operation of SRPS require a trans-disciplinary approach, integrating physical safety, data protection, communication security, human interaction protocols, and ethical considerations, among other aspects.

This research highlights several pivotal findings related to SRPS. Firstly, there is an urgent demand for modernised safety standards specific to SRPS, as existing ones based on industrial robots are inadequate. Data privacy is paramount, necessitating strict adherence to laws like the GDPR and robust encryption. The study stresses the importance of ethical interaction protocols, especially with vulnerable groups, grounded in societal norms and emphasising empathy in robot design. Additionally, SRPS must be environmentally aware and adaptive, ensuring safety and maintaining their integrity. Lastly, the research highlights the growing ethical debates surrounding SRPS, emphasising the need for these robots to uphold human rights and clear ethical guidelines.

This study lays the groundwork for diverse avenues in the domain of SRPS. Areas for exploration include examining the influence of cultural differences on SRPS acceptance, creating standardised testing and validation protocols for SRPS, and delving into real-world case studies to understand practical challenges. Additionally, there is a pressing need to develop a comprehensive security framework for SRPS that addresses cybersecurity, data privacy and ethics concerns [70]. Future research should also assess the legal and ethical regulations governing SRPS, investigate user experience design, and the psychological impacts of SRPS on communities. Such endeavours aim to ensure that social robots are introduced securely and ethically into public spaces, enriching human life while prioritising individual and societal well-being.

Finally, this paper underlines the vital importance of a proactive, integrated, and holistic approach to the security and ethical management of SRPS. As we stand on the cusp of a new era in human–robot interaction, the guidelines and themes highlighted in this study serve as a foundational framework for the responsible development and deployment of these promising technologies. By embracing these guidelines with a commitment to ongoing refinement and adaptability, developers and operators can ensure that SRPS not only enriches our public spaces but does so in a manner that is secure, respectful of individual rights, and aligned with society’s broader safety and well-being.

## Figures and Tables

**Figure 1 sensors-23-08056-f001:**
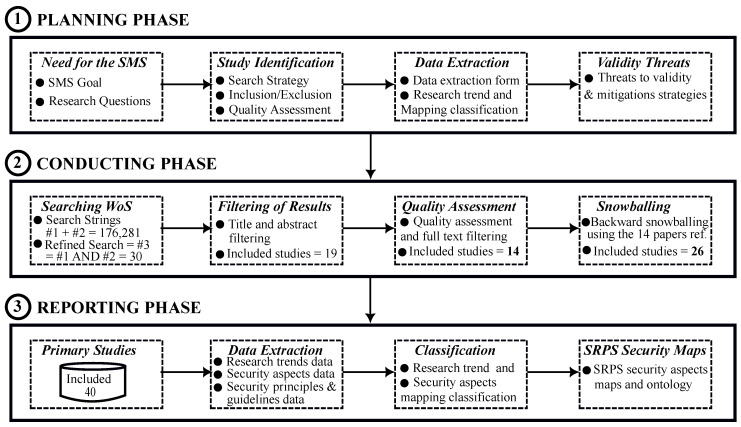
Visual representation of the research procedure for this study.

**Figure 2 sensors-23-08056-f002:**
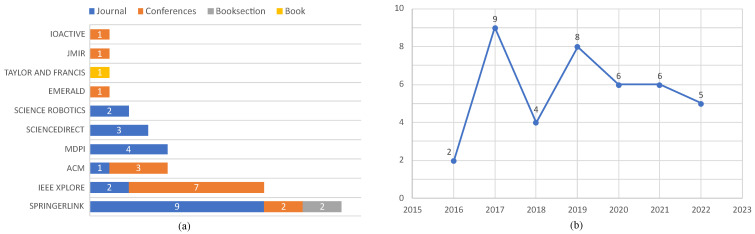
Visual representation of SRPS security aspects research papers. (**a**) Number of and types of papers per digital library, (**b**) number of papers per year.

**Figure 3 sensors-23-08056-f003:**
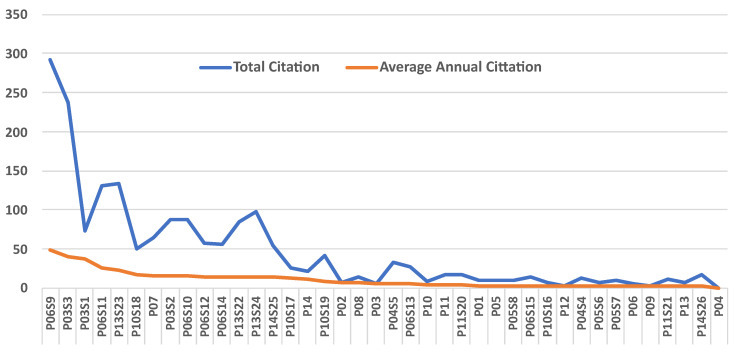
Graphical overview of citation trends in the current study’s research.

**Figure 4 sensors-23-08056-f004:**
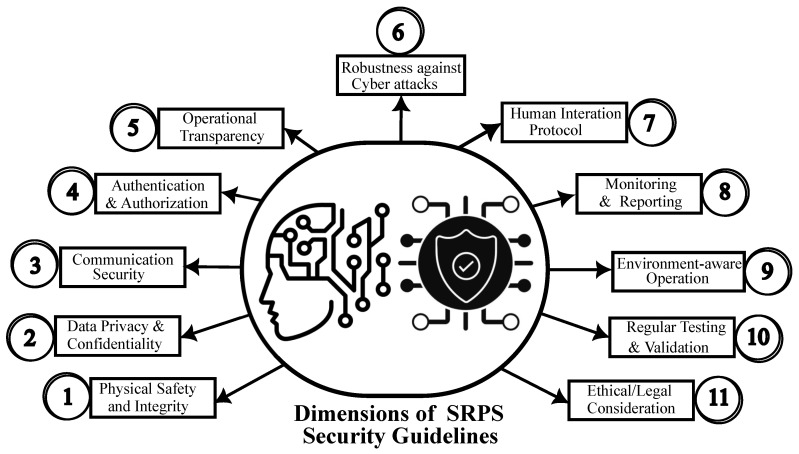
Graphical depiction of proposed security guidelines dimensions for SRPS.

**Figure 5 sensors-23-08056-f005:**
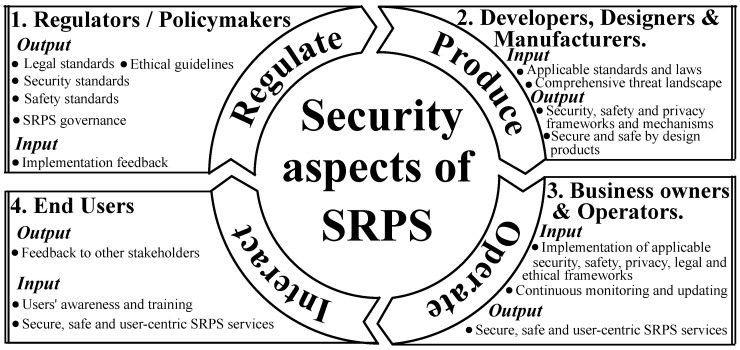
Visual framework illustrating SRPS security dimensions and stakeholder roles.

**Table 1 sensors-23-08056-t001:** Search strings, result counts, and URLs for primary study retrieval.

#	Search String	Results	URL
1	“social robot*”	1874	https://www.webofscience.com/wos/woscc/summary/ca310483-0044-4a22-87bf-b9e7509c4271-9988ee6c/relevance/1 (accessed on 23 July 2023)
2	“*security”	174,251	https://www.webofscience.com/wos/woscc/summary/e076d38e-5e94-49ad-96a2-b1300e5a415c-99890f92/relevance/1 (accessed on 23 July 2023)
3	#1 AND #2	30	https://www.webofscience.com/wos/woscc/summary/ed038b0d-c93f-4778-822e-286eea6badae-99892213/relevance/1 (accessed on 23 July 2023)

**Table 2 sensors-23-08056-t002:** Backward snowballing process summary with reference counts and study IDs.

Study ID	# of Refs.	Round 1	Round 2	Assigned Study ID
P01	37	1	0	
P02	135	0	0	
P03	79	7	3	P03S1, P03S2, P03S3
P04	80	3	2	P04S5, P04S5
P05	48	12	3	P05S6, P05S7, P05S8
P06	58	13	7	P06S9, P06S10 P06S11, P06S12, P06S13, P06S14, P06S15
P07	29	0	0	
P08	110	0	0	
P09	42	5	0	
P10	84	9	4	P10S16, P10S17, P10S18, P10S19
P11	48	2	2	P11S20, P11S21
P12	86	4	0	
P13	46	9	3	P13S22, P13S23, P13S24
P14	111	4	2	P14S25, P14S26
**Total**	**993**	**69**	**26**	

**Table 3 sensors-23-08056-t003:** Temporal distribution and details of primary studies on security aspects of SRPS.

ID	Year	Author	Title
P01	2020	Liu et al. [28]	A Dynamic Behavior Control Framework for Physical Human-Robot Interaction
P02	2022	Farina et al. [29]	AI and society: a virtue ethics approach
P03	2022	Marchang and Di Nuovo [30]	Assistive Multimodal Robotic System (AMRSys): Security and Privacy Issues, Challenges, and Possible Solutions
P03S1	2021	Sharkey and Sharkey [31]	We need to talk about deception in social robotics!
P03S2	2017	Cerrudo [32]	Hacking Robots Before Skynet
P03S3	2017	Wachter et al. [33]	Transparent, explainable, and accountable AI for robotics
P04	2022	Lin et al. [34]	Building a speech recognition system with privacy identification information based on Google Voice for social robots
P04S4	2017	Krupp et al. [35]	Privacy and Telepresence Robotics: What do Non-scientists Think?
P04S5	2017	Rueben et al. [36]	Framing Effects on Privacy Concerns about a Home Telepresence Robot
P05	2020	Abate et al. [37]	Contextual trust model with a humanoid robot defense for attacks to smart eco-systems
P05S6	2019	Silva et al. [38]	Navigation and obstacle avoidance: a case study using Pepper robot
P05S7	2019	Barra et al. [39]	HiMessage: An Interactive Voice Mail System with the Humanoid Robot Pepper
P05S8	2020	Abate et al. [40]	Social Robot Interactions for Social Engineering: Opportunities and Open Issues
P06	2020	Poulsen et al. [41]	Cybersecurity, value sensing robots for LGBTIQ+ elderly, and the need for revised codes of conduct
P06S10	2017	Clark et al. [42]	Cybersecurity issues in robotics
P06S11	2018	Cresswell et al. [43]	Health Care Robotics: Qualitative Exploration of Key Challenges and Future Directions
P06S12	2019	Fosch-Villaronga [44]	Robots, Healthcare, and the Law: Regulating Automation in Personal Care
P06S13	2018	Fosch-Villaronga et al. [45]	Cloud services for robotic nurses? Assessing legal and ethical issues in the use of cloud services for healthcare robots
P06S14	2019	Fosch-Villaronga and Millard [46]	Cloud robotics law and regulation: Challenges in the governance of complex and dynamic cyber–physical ecosystems
P06S15	2018	Poulsen et al. [47]	The Ethics of Inherent Trust in Care Robots for the Elderly
P06S9	2017	Bryson et al. [48]	Of, for, and by the people: the legal lacuna of synthetic persons
P07	2019	Zhang et al. [49]	Emotion-aware multimedia systems security
P08	2021	Saunderson and Nejat [50]	Persuasive robots should avoid authority: The effects of formal and real authority on persuasion in human-robot interaction
P09	2022	Schneider et al. [51]	Stop Ignoring Me! On Fighting the Trivialization of Social Robots in Public Spaces
P10	2021	Giansanti and Gulino [52]	The cybersecurity and the care robots: A viewpoint on the open problems and the perspectives
P10S16	2021	Fosch-Villaronga and Mahler [12]	Cybersecurity, safety and robots: Strengthening the link between cybersecurity and safety in the context of care robots
P10S17	2021	Vulpe et al. [19]	Enabling Security Services in Socially Assistive Robot Scenarios for Healthcare Applications
P10S18	2020	Gordon [53]	Building Moral Robots: Ethical Pitfalls and Challenges
P10S19	2018	Miller et al. [54]	A Case Study on the Cybersecurity of Social Robots
P11	2019	Akalin et al. [55]	The influence of feedback type in robot-assisted training
P11S20	2019	Akalin et al. [56]	Evaluating the Sense of Safety and Security in Human–Robot Interaction with Older People
P11S21	2017	Akalin et al. [57]	An Evaluation Tool of the Effect of Robots in Eldercare on the Sense of Safety and Security
P12	2022	Randall et al. [58]	Top of the class: Mining product characteristics associated with crowdfunding success and failure of home robots
P13	2020	Mazzeo and Staffa [59]	TROS: Protecting humanoids ROS from privileged attackers
P13S22	2017	Breiling et al. [60]	Secure communication for the robot operating system
P13S23	2017	Dieber et al. [61]	Security for the Robot Operating System
P13S24	2016	Dieber et al. [62]	Application-level security for ROS-based applications
P14	2021	Chatterjee et al. [63]	Usage intention of social robots for domestic purpose: From security, privacy, and legal perspectives
P14S25	2019	Chatterjee [64]	Impact of AI regulation on intention to use robots: From citizens and government perspective
P14S26	2016	Pagallo [65]	The Impact of Domestic Robots on Privacy and Data Protection, and the Troubles with Legal Regulation by Design

**Table 4 sensors-23-08056-t004:** Summary of this SMS author affiliation.

Affiliation	Papers	Study IDs
Academia	33	P01, P02, P03, P04, P05, P06, P07, P08, P10, P11, P12, P13, P14, P03S1, P03S3, P04S4, P04S5, P05S6, P05S7, P06S9, P06S10, P06S11, P06S12, P06S13, P06S14, P06S15, P10S16, P10S18, P10S19, P11S20, P11S21, P14S25, and P14S26
Industry	1	P03S2
Mixed	6	P09, P05S8, P10S17, P13S22, P13S23 and P13S24.

**Table 5 sensors-23-08056-t005:** Citation analysis of key studies in SRPS security.

#	ID	Cit${}^{*}$	Avg.C${}^{*}$	#	ID	Cit${}^{*}$	Avg.C${}^{*}$	#	ID	Cit${}^{*}$	Avg.C${}^{*}$	#	ID	Cit${}^{*}$	Avg.C${}^{*}$
1	P06S9	292	49	11	P06S14	56	14	21	P04S5	32	5	31	P12	3	3
2	P03S3	237	40	12	P13S22	85	14	22	P06S13	27	5	32	P04S4	13	2
3	P03S1	73	37	13	P13S24	97	14	23	P10	8	4	33	P05S6	7	2
4	P06S11	130	26	14	P14S25	54	14	24	P11	17	4	34	P05S7	9	2
5	P13S23	133	22	15	P10S17	25	13	25	P11S20	17	4	35	P06	5	2
6	P10S18	50	17	16	P14	21	11	26	P01	9	3	36	P09	2	2
7	P07	64	16	17	P10S19	41	8	27	P05	10	3	37	P11S21	11	2
8	P03S2	88	15	18	P02	7	7	28	P05S8	9	3	38	P13	7	2
9	P06S10	87	15	19	P08	14	7	29	P06S15	14	3	39	P14S26	17	2
10	P06S12	57	14	20	P03	5	5	30	P10S16	6	3	40	P04	0	0

Cit${}^{*}=$ Google Scholar Citation as of 3/08/2023, Avg.C${}^{*}=$ Average annual citation.

**Table 6 sensors-23-08056-t006:** Summary of research types and methodologies employed in our primary studies.

	Categories	Studies	Study IDs
Types	Solution proposal	18	P01, P03, P04, P05, P07, P12, P13, P03S2, P05S6, P05S7, P05S8, P06S10, P06S14, P11S21, P13S22, P13S23, P13S24, and P14S25.
	Philosophical	13	P02, P06, P10, P14, P03S1, P03S3, P06S9, P06S12, P06S13, P06S14, P10S16, P10S18, and P14S26.
	Evaluation	7	P09, P11, P03S2, P04S5, P06S15, P11S20, and P13S23.
	Validation	3	P08, P06S15, and P14S25.
	Experience report	3	P06S11, P10S17, and P10S19.
	Focus group	1	P04S4
Methods	Quantitative	10	P07, P08, P11, P13, P14, P05S7, P11S20, P11S21, P13S22, and P14S25.
	Qualitative	15	P02, P06, P10, P03S1, P03S3, P04S4, P04S5, P06S9, P06S11, P06S12, P06S13, P10S16, P10S18, P10S19, and P14S26.
	Mixed	15	P01, P03, P04, P05, P09, P12, P03S2, P05S6, P05S8, P06S10, P06S14, P06S15, P10S17, P13S23, and P13S24.

**Table 7 sensors-23-08056-t007:** Summary of security aspects covered in our primary studies.

Categories	Studies	Study IDs
Cybersecurity of SR and Users	21	P03, P05, P06, P07, P10, P12, P13, P03S2, P03S3, P04S4, P05S7, P05S8, P06S9, P06S10, P06S13, P10S16, P10S17, P10S19, P13S22, P13S23, P13S24
Data Privacy of Users	20	P02, P03, P04, P05, P06, P07, P12, P13, P14, P03S1, P03S2, P03S3, P04S4, P04S5, P05S7, P06S12, P06S14, P10S17, P14S25, P14S26
Physical Safety of SR and Users	11	P01, P11, P05S6, P06S10, P06S12, P06S13, P10S16, P10S17, P11S20, P11S21, P13S23
Reliability and Continuity of SR	6	P01, P12, P03S2, P06S10, P10S17, P13S23
Legal Framework for SRPS	14	P02, P10, P14, P03S2, P03S3, P06S9, P06S11, P06S13, P06S14, P10S16, P10S17, P13S23, P14S25, P14S26
Ethical consideration for SRPS	16	P02, P06, P09, P10, P03S1, P03S2, P03S3, P04S5, P06S9, P06S11, P06S12, P06S13, P06S15, P10S18, P14S25, P14S26
User influence and manipulation	5	P08, P09, P03S1, P05S8, and P06S13

## Data Availability

The data presented in this study are openly available in Mendeley data at Appendix A.

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
