# Peer review of "Security Aspects of Social Robots in Public Spaces: A Systematic Mapping Study"

_sensors, 2023, doi:10.3390/s23198056_

Round 1

Reviewer 1 Report

The paper proposes a systematic review of security aspects of social robots in public spaces. Specifically it targets a systematic mapping study where the authors review literature from WoS and categorize and synthesize literature on SRPS.

The authors provide a well detailed description of their employed methodology and their achieved results. However, from a technical point of view, the paper is mostly a listing of different research papers. The authors do not explain how this review paper adds value to the existing literature nor provide more detailed and specific challenges in the field of SRPS and potential solutions.

Some recommendations:
- Please clearly state the main contribution of the paper in the introduction section.
- Please explain how this review paper adds value to the existing literature on SRPS. What insights or perspectives does it bring that are not readily available in the individual papers it cites?

- Please emphasize the significance of the review by explaining why a comprehensive understanding of security aspects in SRPS is crucial. How does this knowledge benefit the field of robotics, security, or any relevant application areas?

- Please consider creating a conceptual framework or model to visually represent the key dimensions of security in SRPS and how they interrelate.
- Please create more informative and specific figures that can visually represent the key findings or trends discussed in the paper.
- Please discuss the practical implications of your findings. How can the insights from this review inform the design, deployment, or regulation of social robots in public spaces?
- Please provide more detailed and specific challenges in the field of security in SRPS. You should provide concrete examples and potential solutions. Please elaborate on the implications of these challenges and how addressing them could advance the field.
- Please identify potential future research directions in the field of SRPS security. What areas need further investigation, and what questions remain unanswered?

English is fine

Author Response

Dear Reviewer,

Thank you for your diligent review and the valuable insights and recommendations you have provided to enhance the quality of our paper. We greatly appreciate the time and effort you invested in this process.

We agree with your overall assessment, and your constructive feedback has guided us to improve the manuscript significantly. Incorporating your suggestions has led to a more robust and comprehensive manuscript.

Below, we outline how we have addressed each of your specific recommendations, including placeholder line numbers where the relevant modifications can be found in the revised manuscript:

  1. Stating the Main Contribution in the Introduction (Line 53-84) We have revised the introduction to clearly state the main contributions of our paper, providing readers with a clear understanding of the paper's unique value right from the outset.
  2. Adding Value to Existing Literature (Line 128-166) In this section, we explicate how our review offers new perspectives and insights that are not readily apparent in the individual papers cited, thereby advancing the current understanding of SRPS.
  3. Emphasizing the Significance of the Review (Line 134-166) We have elaborated on why a comprehensive understanding of the security aspects in SRPS is crucial, discussing its benefits to the field of robotics, security, and relevant application areas.
  4. Creation of a Conceptual Framework (Line 864-894) Following your advice, we have developed a conceptual framework to visually illustrate the interrelation between the key dimensions of security in SRPS.
  5. Enhancing Figures to Represent Research Trends Findings (Line 528-529 i.e. Figure 3) We have created more detailed and informative figures that visually represent the pivotal findings and trends discussed in the paper.
  6. Discussion on the Practical Implications of Findings (Line 800-863) We have expanded our discussion to explore how the insights derived from our review can aid in the design, deployment, and regulation of social robots in public spaces.
  7. Detailing Challenges and Solutions in SRPS Security (Line 895-955) This section now articulates the substantial challenges in SRPS security with concrete examples and potential solutions, emphasizing the implications of these challenges and how addressing them could advance the field.
  8. Identifying Future Research Directions (Line 956-997) We have pinpointed potential future research directions in the SRPS security landscape, highlighting areas that necessitate further investigation and posing unanswered questions that represent opportunities for future research.

We are thankful for your constructive feedback, which has been instrumental in elevating the quality of our paper. We look forward to your thoughts on the revisions and remain open to further suggestions.

Thank you once again for your valuable contribution to enhancing our manuscript.

Reviewer 2 Report

The authors present a thorough analysis of works on security aspects of social robots in public spaces. Overall the paper is well written and robust methodologically. The most major concern revolves around the search terms used to identify the body of work used for the study. The authors mention: 

Search String Limitations: The search string might not encompass all relevant studies or may bring in unrelated studies. Mitigation: The search string was iteratively tested and refined to ensure that it’s precise and wide-reaching. Wildcards were used to incorporate variations of the keywords.

That is a bit vague and it is not clear to the reader how the search for the "social robot*" string does not exclude other related works which do not explicitly use this term. There are various "synonyms" for social robots used in related works that could have been excluded through this approach; e.g., works that refer to "robot companions", "humanoid robots", or specific use cases, such as "office robots", "public relation robots", or even refer to relevant research topics, such as human-humanoid interaction (HHI) systems, human/humanoid natural interactions, artificial social intelligence. Have the authors considered and explored these variations in terms of search terms? E.g., did they try but found out it yielded a lot of unrelated studies, so they decided to omit them? This should be further described in the manuscript.

Author Response

Dear Reviewer,

Thank you for your insightful remarks regarding the depth and specificity of the search string utilized in sourcing studies for our systematic review. Your emphasis on the need to consider a wider range of terminologies, including but not limited to “robot companions,” “humanoid robots,” and “office robots,” is well received.

Initially, during the formulation of our research methodology, we recognized the challenge associated with capturing all pertinent research given the variety of terminologies used in different studies to refer to social robots in public spaces (SRPS). We aimed to create a search string that was both inclusive of a wide array of potential terms and focused enough to maintain a scope relevant to our research question.

In our initial searches, we indeed explored the inclusion of a broader set of keywords, which include terms like "human-humanoid interaction (HHI) systems," "human/humanoid natural interactions," and "artificial social intelligence." However, it was observed that expanding the search string to incorporate these additional terms brought in a substantial number of unrelated studies, which did not align with the central focus of our research – the security aspects of SRPS. It was a deliberate decision to fine-tune the search string to a more narrowed focus on source studies most pertinent to the security landscape of SRPS without venturing too far into broader or loosely connected domains.

Furthermore, we initiated an iterative process of testing and refining the search string, involving multiple trial searches and analysis of the results to ascertain the precision and reach of our search strategy. The wildcard “social robot*” was employed to be inclusive of the variations and derivatives of the term without diverging significantly from our core research theme.

Despite the narrowed focus of our search string, we believe that our approach managed to capture a considerable volume of critical research in the SRPS security domain. Nevertheless, we acknowledge that the search string limitation is a valid concern, and it indeed might have led to the exclusion of some pertinent studies using alternative terminologies. We appreciate your suggestion and, recognizing the dynamic nature of terminology in this field, we recommend future reviews in this domain to consider an even more inclusive approach to incorporate the evolving lexicon in the realm of robotics.

To ensure transparency and a clear delineation of our methodology, we have enhanced the section in the manuscript describing the search string development process, including a more detailed account of the iterations and the rationale behind the final choice of terms (Please refer to line 338 - 356 of the updated manuscript under the subheading "Addressing Search String Specificity and Incorporating varied Terminologies".

We are thankful for your constructive feedback, which has been instrumental in elevating the quality of our paper. We look forward to your thoughts on the revisions and remain open to further suggestions.

Thank you once again for your valuable contribution to enhancing our manuscript.
